# Molecular Linking Selectivity on Self-Assembled Metal-Semiconductor Nano-Hybrid Systems

**DOI:** 10.3390/nano10071378

**Published:** 2020-07-15

**Authors:** Thinh Luong The Nguyen, Alba Gascón Nicolás, Tomas Edvinsson, Jie Meng, Kaibo Zheng, Mohamed Abdellah, Jacinto Sá

**Affiliations:** 1Department of Chemistry—Ångström Laboratory, Uppsala University, P.O. Box 532, 751 20 Uppsala, Sweden; luongthethinh.nguyen.9036@student.uu.se (T.L.T.N.); albagasconnicolas@gmail.com (A.G.N.); 2Department of Materials Science and Engineering—Solid State Physics, Uppsala University, P.O. Box 35, 751 03 Uppsala, Sweden; tomas.edvinsson@angstrom.uu.se; 3Department of Chemistry, Technical University of Denmark, DK-2800 Kongens Lyngby, Denmark; jmen@kemi.dtu.dk (J.M.); kaibo.zheng@chemphys.lu.se (K.Z.); 4Chemical Physics and NanoLund, Lund University, P.O. Box 124, 22100 Lund, Sweden; 5Department of Chemistry, Qena Faculty of Science, South Valley University, 83523 Qena, Egypt; 6Peafowl Solar Power AB, Henry Säldes väg 10, 756 43 Uppsala, Sweden; 7Institute of Physical Chemistry, Polish Academy of Sciences, 01-224 Warsaw, Poland

**Keywords:** nano-hybrid systems, self-assembly, functional groups selectivity, spectroscopy

## Abstract

Plasmonics nanoparticles gained prominence in the last decade in fields of photonics, solar energy conversion and catalysis. It has been shown that anchoring the plasmonics nanoparticles on semiconductors via a molecular linker reduces band bending and increases hot carriers’ lifetime, which is essential for the development of efficient photovoltaic devices and photocatalytic systems. Aminobenzoic acid is a commonly used linker to connect the plasmonic metal to an oxide-based semiconductor. The coordination to the oxide was established to occur via the carboxylic functional group, however, it remains unclear what type of coordination that is established with the metal site. Herein, it is demonstrated that metal is covalently bonded to the linker via the amino group, as supported by Surface-Enhanced Resonant Raman and infrared spectroscopies. The covalent linkage increases significantly the amount of silver grafted, resulting in an improvement of the system catalytic proficiency in the 4-nitrophenol (4-NP) photoreduction.

## 1. Introduction

Light conversion into electric and chemical energy is considered one of the key enablers for mankind to switch fossil fuel economy into a green and sustainable economy. Plasmonic metals have optical absorptions that exceed by tenfold their geometric sizes, making their ability to convert light into charge carriers unmatchable. Excitation of their localized surface plasmon (LSP) creates hot carriers [1,2] that can be used in photocatalysis [3,4,5,6,7,8] and photovoltaics [9,10,11]. The direct utilization of hot carriers is difficult due to their ultrafast relaxation and recombination [1]. A common strategy is to transfer the hot carriers to semiconductors, which prolongs the lifetime of the charge-separated state [2,12,13]. However, a direct connection between metal plasmonic and semiconductor results in band bending and higher rates of recombination.

Nano-hybrid systems composed of plasmonic metallic particles and semiconductor oxides connected via a molecular linker demonstrate superior performance in photovoltaic devices [14] and photocatalytic systems [5,6,15]. The linker works primarily as a physical separator preventing the equilibration of the Fermi levels, essential to preserve open-circuit voltage in photovoltaic and redox potential driving force in photocatalytic systems. Additionally, the use of conductive linkers enables precise tuning of open-circuit voltage based on their dipole moment [14] and higher electron injection rates [5].

Furthermore, molecular linkers offer an unprecedented level of system design and tunability because of their controllable ratio between components. The most common discriminator when it comes to molecular linkers is their functional groups. They are often selected to have specific selectivity to metal or oxides. Commonly, functionalities such as carboxylic, phosphate, catechol and silatrane groups bind selectively to oxides, while amino and thiols bind to metal surfaces. The exact nature of the interaction between the functional groups and the specific surfaces remains unsolved, in particular when amino groups are used to coordinate to metallic nanoparticles (NPs), such as Au, Ag and Cu.

The coordination to the oxide was established to occur via the carboxylic functional group, however, it remains unclear what type of coordination is established with the metal site. Herein, a Surface-Enhanced Resonant Raman Spectroscopy (SERS) on the nature of the interaction of p-aminobenzoic acid (pABA) with Ag and TiO_2_ nanoparticles is reported. The studies revealed pABA bonded covalently to TiO_2_ via the carboxylic group and to Ag via the amino group. The covalent linkage improved significantly the amount of silver grafted and consequently the catalytic proficiency in the 4-nitrophenol (4-NP) photoreduction.

## 2. Experimental

### 2.1. Chemicals

All chemicals are obtained from Sigma-Aldrich and have a purity of ≥99% except TiO_2_ powder (20 nm in diameter, 100% pure) that was obtained from Sachtleben Pigments. The chemicals were used without further purification.

### 2.2. Materials Synthesis

Ag nanoparticles (NPs) were synthesized using a modified version of the Lee–Meisel method [16]. Briefly, 0.8 mL of glycerol was diluted in 7.2 mL of doubled deionized (DD) water. Then, 0.2 M AgNO_3_ (0.1 mL) and 0.2 M sodium citrate (0.5 mL) were added to the mixture and stirred to obtain homogeneity. The resultant solution was transferred to a Monowave glass reactor (Anton Paar) and heated until 95 °C. The reaction was kept at that temperature for 30 min under continuous stirring, after which the reaction was cooled to room temperature. The obtained solution was centrifuged once at 14,800 rpm for 20 min to remove unreacted chemicals. It is worth noticing that subsequent centrifugations led to the removal of capping agents, which reduced the stability of the prepared Ag NPs. The concentration of Ag in the stock Ag-pABA solution (25 mL) was estimated to be 2.8 nM, meaning after centrifugation there was Ag NPs 7.08 × 10^−11^ mol per batch.

pABA was subsequently grated onto Ag NPs surface. The centrifuged Ag NPs were diluted in 25 mL using DD water and added to 25 mL of pABA (0.4 mM) solution. The pABA to Ag ratio was selected based on preliminary tests, which show that this is the highest concentration that can be used without modifying Ag NPs morphology and optical properties.

TiO_2_ powder (20 mg) was added to half a batch of Ag-pABA solution (25 mL) to prepare the complete nano-hybrid system. The mix was then sonicated for 1 h and left overnight until colorless supernatant acquired. Then the obtained system was washed with DD water 3 times, each time with a centrifugation at 5000 rpm in 5 min. The final precipitates (30 mg) were diluted in 30 mL DD water and stored in dark for further uses. The other half of prepared Ag-pABA solution (25 mL) was used for control experiment.

### 2.3. Characterization

UV-Vis was employed to characterize the synthesized materials, namely Ag NPs, Ag-pABA and Ag-pABA-TiO_2_. The setup uses a deuterium–tungsten–halogen light source (DH-2000-BAL, Ocean Optics, Largo, FL, USA), a USB4000 spectrometer (Ocean Optics) and an OceanView UV-Vis application.

Transmission electron microscopy (TEM) measurements were conducted on an FEI Titan Analytical 80-300ST TEM.

Dynamic light scattering (DLS) measurements were performed with a Zetasizer Nano ZS from Malvern Ltd. (Malvern, UK) to obtain qualitative data regarding the dimensions of the particles. Disposable cuvettes were used to measure the samples.

Fourier-transformed infrared (FTIR) spectroscopy analyses were performed on solutions dropped on CaF_2_ windows and dried in vacuum. The FTIR measurements were carried out in transmission mode, with a resolution of 4 cm^−1^ and 16 scans average. The experiments were performed in a Bruker Vertex 70v with OPUS application.

Surface-Enhanced Resonant Raman Spectroscopy (SERS) measurements were carried out for three different samples, namely Ag NPs, pABA, and Ag-pABA. Ag NPs were measured directly from purified synthesis solution. pABA was measured as commercial powder. The Ag-pABA sample was prepared from concentrated Ag NPs (10 mL using DD water) mixed with 10 mL of pABA (1 mM) aqueous solution. The mixture was centrifuged at 14,800 rpm for 20 min and the solid deposit containing the Ag-pABA was used in the measurement. The experiments were performed in a Reinshaw inVia Qontor Raman spectrometer, using 20× objective, a 2400 lines/mm grating, and 405 nm excitation laser.

### 2.4. Catalytic Reduction of 4-Nitrophenol

From the suspension of 30 mg catalyst in 30 mL DD water, the subsequent suspensions with various concentrations (10, 5 and 2.5 μg/mL) were obtained by dilution. The suspensions were degassed for 15 min with argon to remove oxygen. Afterwards, 0.5 mL of degassed 4-nitrophenol (4-NP) (0.2 mM) solution, 0.1 mL of degassed NaBH_4_ (0.1 M) and 0.5 mL of degassed catalyst were added to the 1 cm quartz cuvette. The cuvette was subsequently sealed and reaction started in the presence and absence of light. Catalytic activity was assessed by online UV-Vis spectrometer. The reactions were performed at room temperature and excited with a 405 nm continuous wave laser.

Two control experiments were conducted, using Ag-pABA solution and TiO_2_ suspension. Roughly 2 mg of TiO_2_ powder was loaded into 50 mL of DD water (40 μg/mL) and the suspension stirred. The other control experiment with Ag-pABA was carried out at the dilution ratio used to obtain the lowest catalyst concentration (2.5 μg/mL). From 25 mL of the Ag-pABA solution, 0.125 mL was taken out and diluted to 60 mL by DD water. Subsequent steps were similar to the described catalytic reduction, except for only the experiments with laser excitation were conducted. It is worth mentioning that catalyst amount was chosen to obtain first order reaction, enabling extraction of the reaction rate. Slightly higher catalyst amount led to rapid reagent consumption and variable kinetics.

The reaction can be monitored online via UV-Vis since in the reaction mixture the 4-NP is rapidly converted into 4-nitrophenolate (reaction pH is 9) that has a characteristic absorption at 400 nm and the product 4-aminophenol (4-AP) has an absorption at 300 nm [17,18,19].

## 3. Results and Discussion

The UV-Vis spectrum of citrate-capped Ag NPs (Figure 1) shows a sharp plasmonic resonance centered at 400 nm and a UV peak at below 300 nm. The particles were found to be elongated spheres according to TEM (Figure 2a) and confirmed by the double peak on DLS measurement (Appendix A). If the particles were perfect spheres (classic citrate method), one would expect particles of diameters around 20 nm, based on the UV-Vis absorption [20]. However, the addition of glycerol, as stabilizer [21], contributed to preferential growth of certain phases leading to particle elongation. The average particle size from DLS analysis was estimated to be about 40–50 nm long and 4–6 nm wide. This matches relatively well with TEM analysis (Figure 2a), however, the TEM sizes are a bit smaller than expected since DLS sizes include the capping agent. However, we cannot exclude the presence of small amounts of unreacted Ag NPs seeds as responsible for the DLS signal related to 4–6 nm, which are formed in situ in the first stages of the sample preparation.

The addition of pABA to silver did not significantly affect the morphology of the nanoparticles (Figure 2b and Appendix A). However, it induced a small redshift in Ag absorption (ca. 2 nm), and a blueshift in pABA absorption (ca. 15 nm), which is indicative of strong electronic interaction between Ag NPs and pABA. It is worth mentioning that the measurements were performed at similar pH levels and thus the observed changes in p-ABA peak position are not related to different forms, namely neutral and anionic.

The FTIR spectrum of Ag NPs can be found in Figure 3 (complete mid-IR spectra in Appendix A). The spectrum is dominated by peaks ascribed to glycerol (1121, 1328, 1411, 2878, 2934 and 3261 cm^−1^) [22], and citrate (1376 and 1592 cm^−1^ corresponding to the COO– symmetric and asymmetric stretching, respectively) [23]. The presence of glycerol after washing confirms its involvement in shaping the final morphology of the Ag NPs. Since the peak at 1376 cm^−1^ related to citrate is overlapped by the glycerol signal, a deconvolution was performed to reveal the COO– symmetric stretching of the citrate. The intensity ratio between the symmetric (1376 cm^−1^) and asymmetric (1592 cm^−1^) peaks on sodium citrate is less than one, while on Ag NPs it is roughly three, indicating the linkage of COO– group to Ag (see Appendix A). When citrate binds with Ag, the electrons on COO– are delocalized, creating an ionic-like bond with Ag [24]. Hence, the asymmetric stretching is significantly reduced leading to a higher ratio.

The first evidence for the grafting of pABA on Ag NPs was provided by UV-Vis (Figure 1). The addition of pABA to the Ag NPs solution led to a blue shift of the pABA absorption peak at 280 to 265 nm. This shift was found to be significantly larger than the one observed when pABA was bonded to TiO_2_ (see Appendix A) that is known to occur via the carboxylic group [9], and thus suggested a different binding motif i.e., via the NH_2_ group. Additionally, the absorption peak of citrate-capped Ag NPs shows a slight red shift with the pABA addition, indicating the small size increase in Ag after the binding with pABA.

The linking of pABA and Ag NPs through the amino group was corroborated by the appearance of a peak at 1310 cm^−1^ in the FTIR spectrum of Ag-pABA, assigned to the C–N stretching. This is also accompanied by a decrease in COO– ratio between symmetric and asymmetric stretching compared to Ag NPs. The observations suggest pABA linkage to Ag via the NH_2_ group by replacement of citrate COO– groups, justifying the decrease in particle stability when higher concentrations of pABA are used.

Unfortunately, the disappearance of NH_2_ stretching around 3300 cm^−1^ is not observable in the FTIR spectrum due to the overlap with the strong O–H stretching region. Its disappearance was assessed by SERS. In Figure 4, the peak regarding C–N stretching located at 1289 cm^−1^ in free pABA redshifts by 23 to 1266 cm^−1^ in Ag-pABA. Furthermore, the vibrations of N–H stretching mode at 3364 and 3463 cm^−1^ are absent in the spectrum of Ag-pABA. This evidence confirms covalent bonding of pABA to Ag via the amino group.

Published literature suggested that pABA preferred binding mode on TiO_2_ is the bidentate [25,26,27], a mode involving the bridging of the two oxygen atoms of the carboxylic group neighboring 5-fold-coordinated Ti atoms. As a consequence, one should expect the disappearance of C=O and O–H vibrations as pABA binds to TiO_2_. Figure 5 shows the FTIR spectrum of pABA on TiO_2_. The vibrational signal at 1665 cm^−1^ related to C=O is completely absent, accompanied by the disappearance of the O–H bending peak at 1422 cm^−1^. Additionally, also visible in the spectra were the peaks of N–H symmetric and asymmetric stretching located at 3364 and 3460 cm^−1^, respectively. However, the presence of O–H stretching in this region partly omits the N–H vibrations. This O–H group originates from the absorption of water vapor in the atmosphere on TiO_2_, which occupies a large number of binding sites on the semiconductor.

In the complete system where citrate ions present, it is hard to observe the individual disappearance of C=O and O–H due to the size of the citrate shell. The particle size of Ag is comparable to that of TiO_2_, which means a large number of citrate ions can be found surrounding Ag NPs. However, the drop in the intensity of both COO– symmetric and asymmetric stretching modes in Figure 6 (1376 and 1603 cm^−1^, respectively) is strong evidence for Ag loading on TiO_2_ via the pABA carboxylic groups. In theory, the COO– of citrate on Ag can link directly to TiO_2_ but that did not happen even after 48 h due to surface charge repulsion (both negative).

To support the linkage of TiO_2_ to the Ag-pABA, UV-Vis measurements were performed (Figure 7). The spectrum related to the full system shows clearly the absorptions of the three species. It is worth noticing that, their absorptions shift once more indicative of electronic coupling between the components.

The 6e^−^ catalytic reduction of 4-NP was used as a model reaction to test the photocatalytic performance of the systems since this is a trusted reaction to test nanoparticles reactivity [28] and plasmonic photocatalysis [29,30]. The mechanism of the reduction has been studied thoroughly elsewhere [31,32], in which the surface of transition metal is a favorable binding site of 4-NP. In the presence of NaBH_4_, the reduction follows the Langmuir–Hinshelwood mechanism, which involves the reaction of the active hydrogen (hydride) from NaBH_4_ produced on the metal with the adsorbed 4-NP. In a classical catalytic reduction, the reaction is limited by amount hydride at the reaction site. This limitation can be significantly improved in a photocatalytic system where electrons are supplied both from hydride and photo-process. The latter uses solution protons to form the products, namely one 4-AP and two water molecules per 4-NP.

When Ag-pABA-TiO_2_ is employed as the reduction catalyst (no light), the system exhibits a linear conversion of 4-NP with a depletion rate of −8.0 × 10^−5^ min^−1^ (Figure 8). The substrate was converted solely to 4-AP as confirmed by the isosbestic point in the UV-Vis spectra (see Appendix A). Similar behavior was observed when the reaction was carried out in the presence of light, however, the depletion rate increased by 7.4 times (rate = −5.9 × 10^−4^ min^−1^) (Figure 8). It should be mentioned that under the reactions conditions there was no detected activity when Ag and/or TiO_2_ were absent. Experiments with TiO_2_ showed a drop a small drop in 4-NP concentration in solution after 60 min (see Appendix A), which was found independent of illumination. Since 4-AP was not detected, the drops were suspected to be related to adsorption and not to photo-reactivity.

It is worth mentioning that reactions were carried out with low amounts of silver, to extract kinetic rates. Since the average particle sizes of Ag and TiO_2_ are not too dissimilar, one can estimate their ration using the weighted content of each component. Under reaction conditions utilized, there were about 10^7^ times more TiO_2_ particles than Ag, meaning that a large amount of TiO_2_ did not have silver and consequently the ones that have silver should be on a ratio 1:1.

Figure 9 shows that higher amount of silver resulted in faster catalysis. Metal nanoparticles are also catalytically active, even in the absence of TiO_2_ and/or light [26]. Therefore, catalytic activity studies on Ag-pABA were performed to establish their contribution to overall activity. As expected, silver shows some activity (Figure 9) that follows the reactivity obtained with the complete system in the absence of light. Since light did not affect the catalytic activity of Ag-pABA, it is reasonable to assume that Ag plasmon hot electrons in the absence of TiO_2_ do not participate in the reaction, and this consequently demonstrates that the enhancement observed in the complete system in the presence of light is due to plasmon hot electrons. The latency observed for Ag-pABA-TiO_2_ (10 μg/mL) seems to be related to a small change in the pH of the solution that stabilizes NaBH_4_, resulting in a delayed start of reaction as observed elsewhere [18]. This further supports the strategy to extract kinetic data with low silver concentration and as fresh as possible NaBH_4_ solutions.

## 4. Conclusions

In this study, the self-assembly linkage motifs of Ag NPs and TiO_2_ with a molecular linker (pABA) have been successfully investigated by FTIR and Raman. pABA binds to the surface of Ag NPs through the amino group, and the surface of TiO_2_ by the carboxylic end. The coordinated system was found active in the reduction of 4-NP. The rate of reaction increased significantly when the system was illuminated, which demonstrates direct involvement of Ag plasmon hot electrons in the reaction. pABA is the key component in the system which quickly transfers the high energy electrons from Ag NPs to the conduction band of TiO_2_.

## Figures and Tables

**Figure 1 nanomaterials-10-01378-f001:**
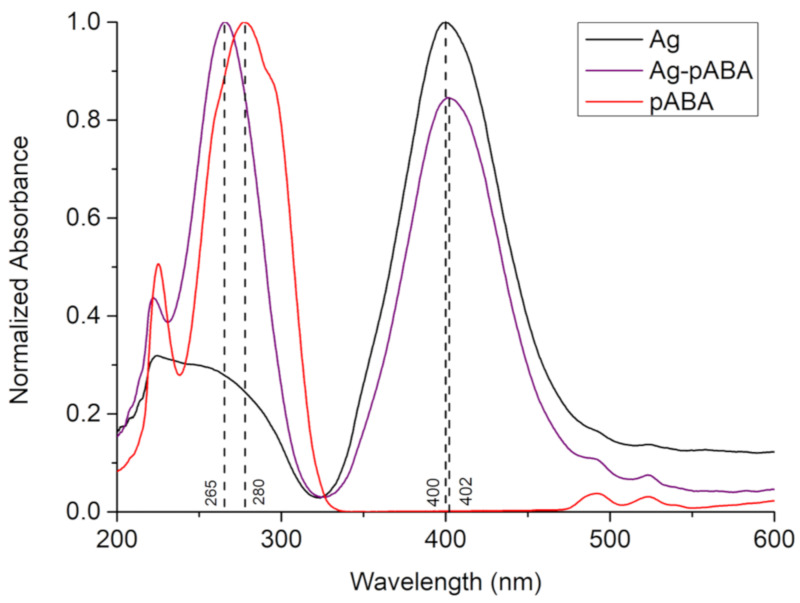
UV-Vis absorption spectrum of Ag (black trace), p-aminobenzoic acid (pABA) (red trace), and Ag-pABA (purple trace).

**Figure 2 nanomaterials-10-01378-f002:**
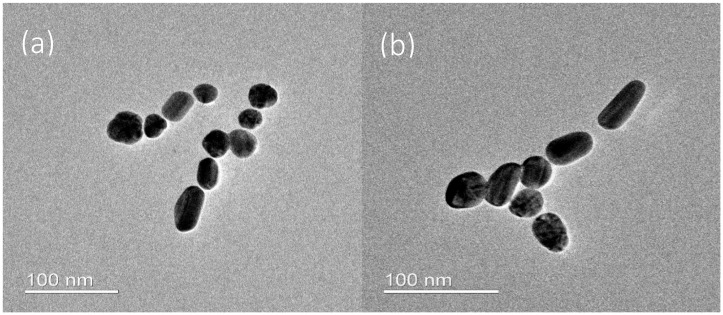
Transmission electron microscopy (TEM) micrographs of (**a**) Ag nanoparticles (NPs), and (**b**) Ag-pABA.

**Figure 3 nanomaterials-10-01378-f003:**
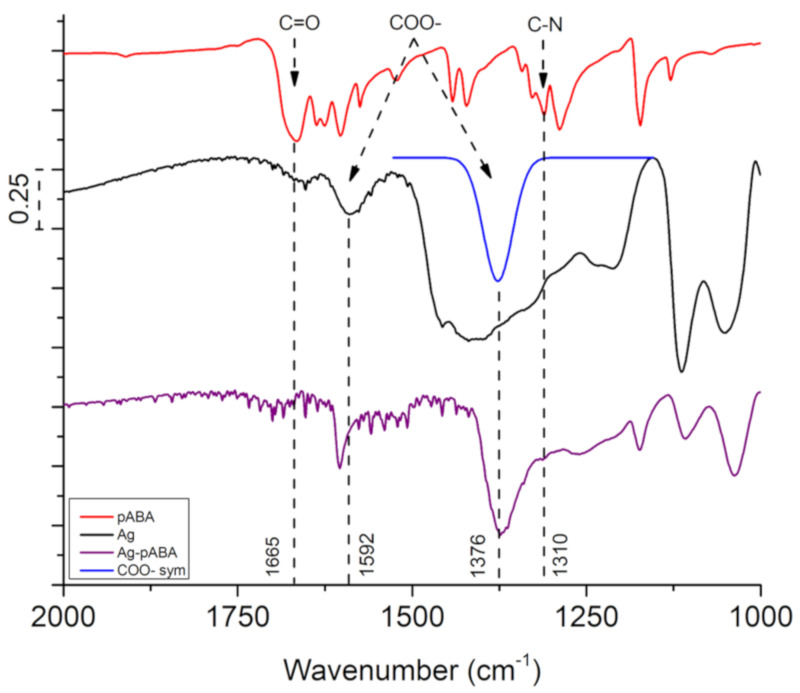
Fourier-transformed infrared (FTIR) spectra of pABA (red trace), Ag NPs (black trace), and Ag-pABA (purple trace). The fit peak (blue trace) was obtained from deconvolution, showing COO—symmetric stretching.

**Figure 4 nanomaterials-10-01378-f004:**
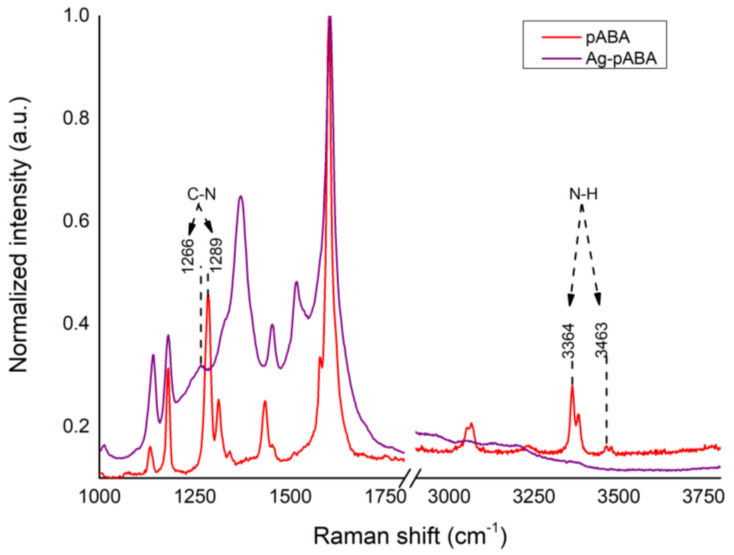
Raman analysis using 405 nm excitation laser. Non-resonant Raman spectrum of pABA (red trace), and Surface-Enhanced Resonant Raman Spectroscopy (SERS) spectrum of Ag-pABA (purple trace).

**Figure 5 nanomaterials-10-01378-f005:**
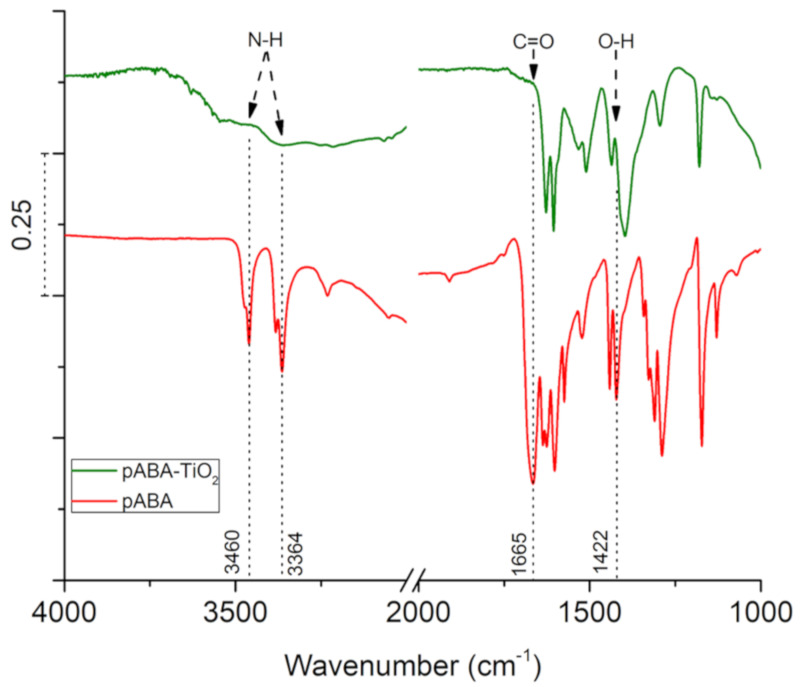
FTIR spectra of pABA (red trace), and pABA-TiO_2_ (green trace).

**Figure 6 nanomaterials-10-01378-f006:**
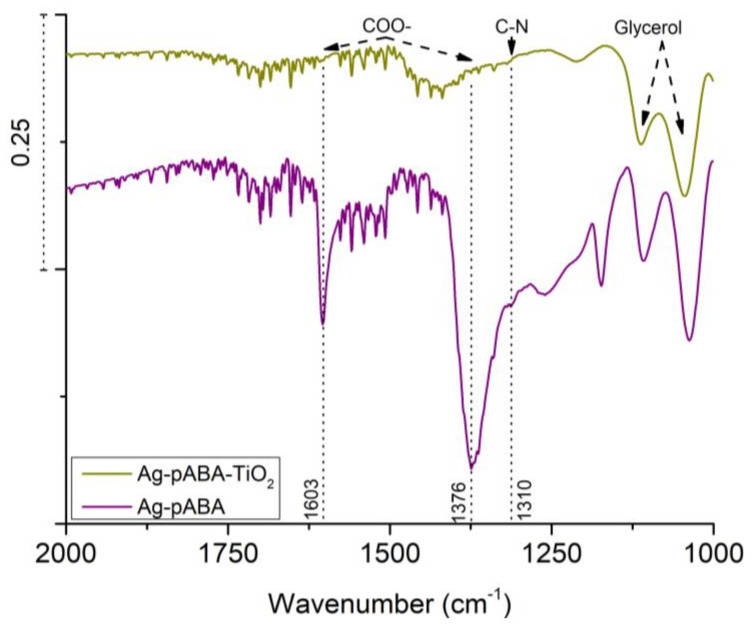
FTIR spectra of Ag-pABA-TiO_2_ (dark yellow trace) and Ag-pABA (purple trace).

**Figure 7 nanomaterials-10-01378-f007:**
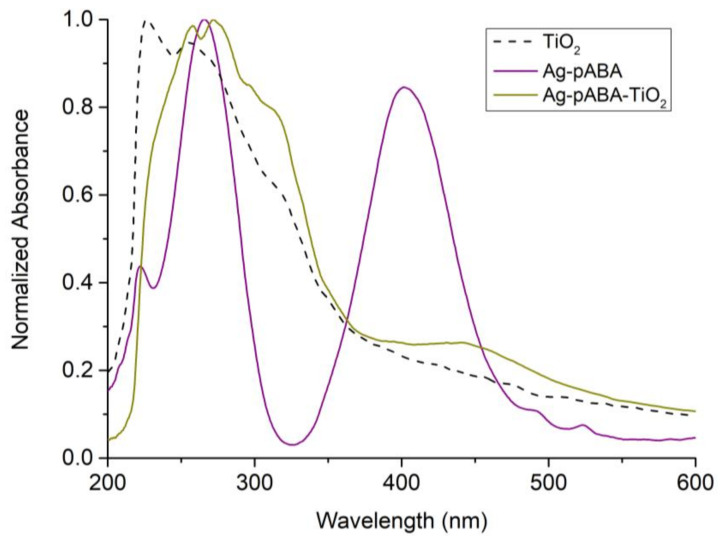
UV-Vis absorption spectrum of TiO_2_ (black dashed trace), Ag-pABA (purple trace), and Ag-pABA-TiO_2_ (dark yellow trace).

**Figure 8 nanomaterials-10-01378-f008:**
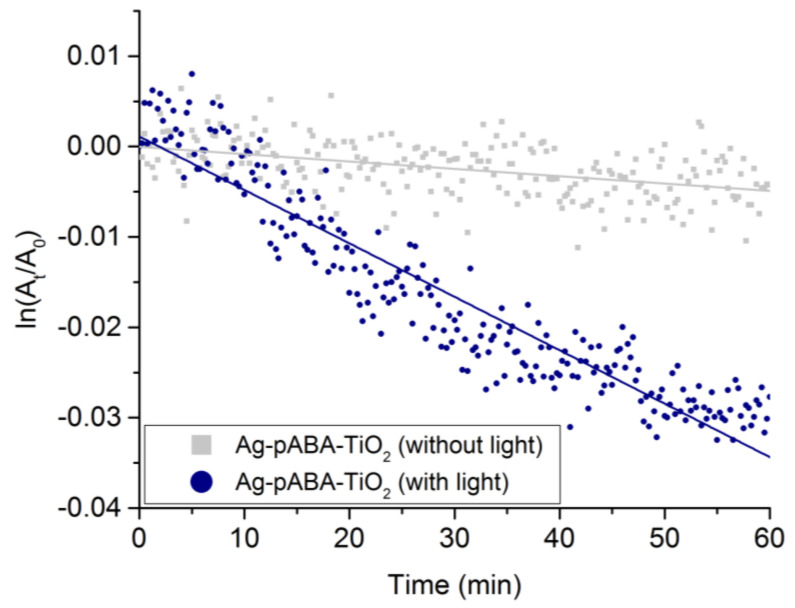
The disappearance rate of 4-nitrophenol (4-NP) in Ag-pABA-TiO_2_ (catalyst amount of 2.5 μg/mL), with and without laser excitation.

**Figure 9 nanomaterials-10-01378-f009:**
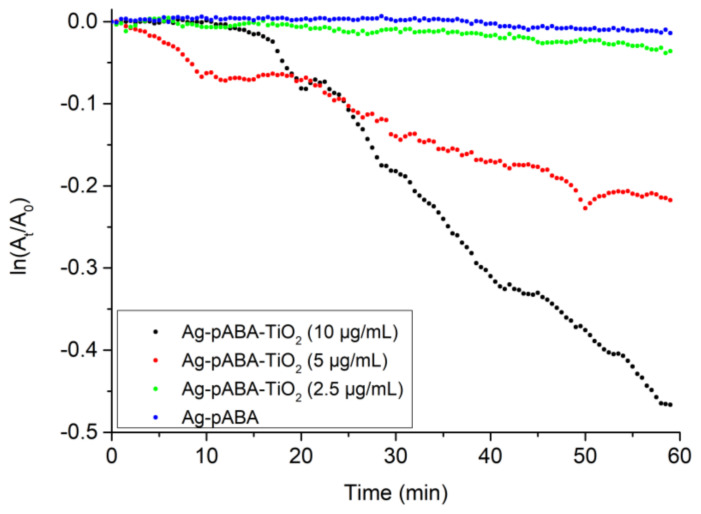
The disappearance rate of 4-NP in Ag-pABA-TiO_2_ with different concentrations of Ag-pABA (Ag-pABA amounts under parenthesis). For comparison purposes, the activity of Ag-pABA at similar concentration as the lowest (2.5 μg/mL) amount of catalyst is also plotted.

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
