# Peer review of "Molecular Linking Selectivity on Self-Assembled Metal-Semiconductor Nano-Hybrid Systems"

_nanomaterials, 2020, doi:10.3390/nano10071378_

Round 1
Reviewer 1 Report
The authors present spectroscopic results to elucidate the binding mode of the molecule pABA to the Ag nanoparticle surface, and then link this to TiO2 supports for plasmon-driven photocatalysis of 4-NP to 4-AP. The study is well performed and the conclusions are substantiated by the data. The presentation of some of the data, however, has made it difficult to easily compare the data with the discussion. On several occasions, I found myself needing to refer to the spectra in the SI figures to convince myself that the peak assignments were correctly referenced. This made it difficult to follow the discussion of these data in the manuscript because I was constantly going back and forth between the figures. I recommend accepting this manuscript for publication after the following minor revisions are noted:
- The authors should consider revising their presentation of the spectra shown in Figure 1. Although the use of the red and black color for the assignment of the various peaks to different parts of the sample (the ligands, red) and the Ag NPs (black) was a nice way to show the composite system, this was also somewhat puzzling at first glance. I needed to lo atok the SI to find the spectra shown in Figure S1 and Figure S3 to see the spectra of just the Ag NP and just the pABA molecule. I would suggest to simply plot the spectra shown in Figure S1, Figure S3 (purple curve) with the combined Ag-pABA spectra in Figure 1. That would make the whole peak assignment much more obvious to the reader.
- The discussion of the FTIR results are difficult to follow with regards to the discussion of all the peaks in the spectra. The authors are encouraged to show some of the spectra contained in the Supporting Information into the main text figures. In particular, the surface coverage of stabilizing surfactants/ligands on the Ag surface is very complex due to both glycerol and citrate from the synthesis. Without referencing the SI data Figure S2, the reviewer was at first unconvinced by the assignment of certain peaks. I suggest to address this in one of two ways: either
- Add the glycerol and citrate spectra into the figure, which may crowd the figure considerably, so the authors may also consider to
- Use symbols (e.g. *) to highlight peaks in the spectra that belong to either of these two molecules, which are not central to the main story.
- The authors could also consider showing the full spectra in the SI, and instead of only showing certain spectral regions that are needed to draw their conclusions. For instance, the broad spectral region from 4000 – 3000 cm-1 contains little information due to the broad OH stretch, as noted from the authors, so this may be omitted to focus on the region from ~1700 – 1000 cm-1 in Figure 3.
The rest of the figures are easy to follow, so if the others are slightly altered the manuscript overall will be improved.
There are also several grammatical errors that should be corrected throughout. Some of the noticed errors include:
The abstract, should read “Aminobenzoic acid is a commonly…to connect the plasmonic metal to an oxide-based semiconductor.” “What type of coordination…” “…it is established that the metal…” “…silver that can be grafted, resulting…”
Page 1, line 36: should read “plasmonic metal”
Page 2, line 46: should read “controllable ratio between components.”
Page 2, line 54: should read “what type of coordination…”
There are several other instances throughout the text that I cannot list explicitly line by line. The authors should do another grammatical check of their manuscript prior to submission.
Author Response
We would like to thank the Reviewer for the time dedicated to the revision of the manuscript and valuable comments. Please find the answer to the comments in the attached file. The manuscript and SI were revised accordingly.

Reviewer 2 Report
The present manuscript describes the nano-hybrid systems and their selective self-assembly. The new system proved to be efficient as a catalyst in the photoreduction reaction. Overall, the study is nice and simple and fits into the theme of the journal but it lacks structural characterization of the entire hybrid system. Moreover, the figures could use some care and complexity. Therefore, I would recommend the publication of the manuscript after major revisions.
- The chemical list has to be updated with the purity of each chemical or further details if needed.
- Figure 1 can be enriched with the abs spectra of the Ag NPs before the addition of p-aba and after, the spectrum of the p-aba alone.
- Figure 2 doesn’t look representative to me. Could the autors add a HRTEM and a low resolution TEM for each case showing the narrow size distribution? These images show a rather polydisperse assemble, which can be explained by the used synthetic method (microwave). In this case, the authors should not claim narrow size distribution but focus on the other advantages of the NPs.
- The authors describe a shift in the absorption peak of the Ag NPs but it is not clear from the way how they presented the data (please see suggestion point 1)
- The authors mentioned that the first suggestion of the grafting of pABA on Ag NPs is the additional peak in absorption but that can also show up if you mix the 2 compounds. A clearer analysis and explanations has to be given.
- The authors should work out the data in figure 3 showing what vibration correspond to what functional group, at least the ones of interest.
- Comment 6 applied also for figure 4, 5 and 6.
- The authors should give TEM for the entire hybrid system including the TiO2 NPs or other structural characterization proving the the TiO2 is actually there and bound the ligand.
- Did the authors perform a control experiment of the catalytic reaction with and without Ag and second one, with and without TiO2?
Author Response

(The authors gave the same response as above.)

Reviewer 3 Report
The manuscript describes the synthesis of nano-hybrid systems composed of silver (Ag) and titanium oxide (TiO2) nanoparticles (NPs) linked by para-aminobenzoic acid (pABA). FTIR and Raman spectroscopies are used to investigate the connection between the NPs and the linkers and the formed systems are tested for the catalytic reduction of 4-nitrophenol (4-NP). However, as it is, I do not recommend the manuscript for publication. The main remarks are the following.
Experimental part
-line 77: what is the amount of TiO2 powder added to the Ag-pABA solution to form to hybrid system?
-lines 94-95: as written, it is not clear if the Ag-pABA samples for the SERS experiments are prepared with the same ratio of pABA to Ag as the others. However, the authors mention that this ratio is important to control the Ag NPs morphology and optical properties (lines 75-76).
-line 111: The authors indicate that in the reaction mixture the 4-NP is rapidly converted into nitrophenolate; what is the pH of the different solutions?
Results and discussion
Concerning the UV-visible absorption spectra, the spectra of Ag NP and pABA should be given in the article, not in SI, and presented in the same figure as the spectrum of Ag-pABA. That would help to evidence the shift in the pABA absorption peak as said line 140. However, the spectrum of Ag NPs (Fig. S1) must be checked because it should not have a zero absorbance at 320 nm; maybe a mistake in background correction.
The authors consider that they obtain a narrow particles size distribution from the absorption spectrum, but on the TEM image the particles are not monodisperse. Moreover, the particles are not spherical but elongated (Fig. 2a) and that seems even more pronounced after addition of pABA (Fig. 2b) while the authors consider no effect of pABA on the morphology (line 128).
The analysis of the FTIR spectra is not clear and convincing.
- The deconvolution used to reveal the COO- symmetric stretching of the citrate at 1376 cm-1 should be detailed as the intensity ratio of the symmetric and asymmetric stretching peaks is used to corroborate the linkage of COO- to Ag (Fig.3). This ratio is difficult to evaluate for citrate alone as the FTIR spectrum of sodium citrate given in Fig.S2 does not show clearly the COO- stretching peaks at 1376 and 1592 cm-1. This ratio is also used to suggest the linkage of pABA by NH2 group.
- Afterwards, the binding of pABA to TiO2 is inferred by the absence of the C=O and O-H vibrational signals at 1665 and 1422 cm-1 (Fig. 6). But, these signals were also absent in the spectrum of Ag-pABA.
- For the total Ag-pABA-TiO2 system, the FTIR spectrum is not well resolved and seem dominated by glycerol peaks.
Regarding the Ag-pABA-TiO2 system, what is the proportion between Ag-pABA NPs and TiO2 NPs in solution and linked together? What is the number of Ag-pABA NP and TiO2 NP in the final system? One and one? TEM images of the total Ag-pABA-TiO2 system should be provided.
Concerning the catalytic test, there is no clear evidence that the 4-NP reduction reaction occurs. In the absence of light, the spectra are almost all identical during the reaction time of 1 hour. What is rather surprising because metal nanoparticles are known to catalyse the reduction of 4-NP by NaBH4 (see for instance, T. Aditya, A. Pal and T. Pal, Chem. Commun., 2015, 51, 9410-9431.); so, here, Ag NPs seem to have no catalytic effect. under irradiation, a small decrease in the absorbance of 4-NP is observed (around 4%) after 1 hour but the presence of an isosbestic point is not obvious. To show the reaction with a noticeable conversion ratio, spectra on a longer reaction time should be presented.
Author Response

(The authors gave the same response as above.)

Reviewer 4 Report
Line 121: can the authors provide the mean size and dispersion in their Ag colloids by using TEM images? Instead of qualitatively saying that a narrow plasmon peak means a narrow size distribution.
Line: 142: there is a reference error/missing
The FTIR part is very interesting and nice. I think it is the highlight of the paper and it is very nicely described.
Line 145: Why Ag nanoparticles would growth after adding pABA? This is not clear to me.
Can the authors add in Figure 5 the citrate infrared spectra for comparison as well?
Can the authors show some TEM images after the coupling between Ag nanoparticles and TiO2 powder? How does it look like? Which is the density of the coupling?
Key references missing. The authors should carefully check the literature. In my opinion there are many key references missing, here I include some:
Plasmon-semiconductor interface for photocatalysis: Science 2015, 349 (6248), 632-635
Line 200/201: NTP to ATP conversion on Ag particles involving hot electrons: Nat. Commun. 2017, 8, 14880
Recent advances on plasmonic photocatalysis (and reference therein), for example Acc. Chem. Res. 2019, 52, 9, 2525–2535.
Author Response

(The authors gave the same response as above.)

Round 2
Reviewer 2 Report
I would like to thank the authors for addressing the questions and to suggest the publication of the manuscript as is.
Author Response
We thank the Reviewer for the time dedicated to the revision of the manuscript and the endorsement for publication.
Reviewer 3 Report
The manuscript has been improved compared to the first version, the figures are clearer. However, even if the authors have provided more data and analysis of their results, the given explanations are not always clear, appropriate and convincing. Therefore, I do not recommend the manuscript for publication. The main remarks are the following.
Experimental part
-The authors have indicated the amount of TiO2 powder added to the Ag-pABA solution, but they should also have indicated the amount of the initial Ag NPs they obtained after the centrifugation.
-For the control experiments with TiO2 suspension, the used amount of TiO2 (40 µg/mL) is much higher compared to the concentrations of catalysts (10 µg/mL max.); it should have been adjusted as it was done for Ag-pABA solution, even if in that case the choice of the lowest concentration (2.5 µm/mL) is debatable.
-The catalytic reaction is monitored spectrophotometrically; the authors indicate the characteristic absorption wavelengths of the reactant and the product and they have added a reference but the given reference (DOI: 10.1021/acs.jpcc.9b07114) is not appropriate here. Indeed, the reference does not present the absorption spectra of the compounds, but in the conclusion the authors write “we recommend performing the catalytic reduction of 4-NP in aqueous NaBH4 solution at pH 13 by controlling the pH via the addition of a strong base like NaOH” while in the present study the reaction pH is 9.
Results and discussion
The authors have modified Figure 1 according to the remarks and have added a small paragraph to describe it. Regarding the absorption spectrum of Ag NPs, their analysis is incorrect: the second peak below 300 nm for the Ag NPs is not indicative of elongated NPs. The peak below 300 nm is always present whatever the shape pf the Ag NPs. Elongation of spherical NPs induces a red shift of the main plasmon resonance band around 400 nm and for nanorods a second band is observed in the red (its position depends on the aspect ratio).
In Figure 1 the authors also present the absorption spectra of pABA alone and in the presence of Ag-NPs and they observe “a blueshift in pABA absorption (ca. 15 nm), which is indicative of a strong electronic interaction between Ag NPS and pABA”. The authors should be more careful in their affirmation and check the pH of the two solutions as the neutral form of pABA absorbs around 275 nm while the anionic form absorbs around 260 nm. Moreover, the absorption spectrum of pABA given in Figure S4 differs from that in Figure 1, so how to compare the shift upon addition of TiO2 or Ag nanoparticles?
In addition to TEM images, the authors have added DLS measurements to confirm elongated shapes of the NPs. From the DLS analysis showing two peaks, the first around 5 nm and the second around 40 nm, the authors conclude that “The average particle size from DLS analysis was estimated to be about 40-50 nm long and 4-6 nm width. This matches relatively well with TEM analysis (Fig. 2a)”. However, on the TEM image shown, the NPs have a width of at least 20 nm. So, the DLS peak at 4 nm cannot be attributed to the shown Ag-NPs.
The new Figure 3 with the assignments of the peaks and the change of the citrate FTIR spectrum in Figure S3 make the discussion easier to follow, but the analysis of the FTIR spectra is still not convincing. In particular, it is difficult to use the intensity ratio between the COO-symmetric (1376 cm-1) and asymmetric (1592 cm-1) stretching peaks to show the linkage of citrate by the COO- group on Ag. Indeed, the COO- symmetric peak overlaps with the glycerol peaks, so the ‘original peak intensity’ at 1376 cm-1 cannot be assigned to the intensity of the COO- symmetric stretching only.
Then, the authors indicate “the linking of pABA and Ag NPs through amino group was corroborated by the appearance of a peak at 1310 cm-1 in the FTIR spectrum og Ag-pABA, assigned to the C-N stretching.” This tiny peak indicates the presence pABA on the Ag NPs but not the way it is linked. They also invoke “a decrease in COO- ratio between symmetric and asymmetric stretching compared to Ag NPs”, but this COO- ratio is related to citrate not pABA and for Ag-pABA the contribution of glycerol is much smaller compared to Ag NPs, inducing a decrease in the intensity at 1376 cm-1.
Afterwards, for the total Ag-pABA-TiO2 system, the authors write “the drop in the intensity of both COO- symmetric and asymmetric stretching modes in Figure 6 (1376 and 1603 cm-1, respectively) is strong evidence of the linkage of pABA molecule with TiO2 through the carboxylic group”. But, these peaks are related to citrate ions not pABA, so I do not follow the reasoning. Moreover, I wonder what would have been the result of a mixture of TiO2 and Ag NPs (without pABA).
To support the linkage of TiO2 to Ag-pABA, the authors have added the UV-vis absorption spectrum of the entire system, but TEM images showing the coupling between TiO2 and Ag are still missing as the NPs have similar sizes. In the catalysis part, the authors consider that as” there were about 1000 times more TiO2 particles than Ag, a large amount of TiO2 did not have silver and that the ones that have silver should be on a ratio 1:1”, but they have no proof. Moreover, I do not understand how the 1000 factor is obtained using the weighted content of each component.
Concerning the catalytic test, the new figure S6 clearly shows the 4-NP reduction reaction with the formation of 4-AP, but for that, the amount of catalyst has been increased. The authors justify the use of low amounts of catalysts to extract kinetics, but even for a complex reaction, it is always possible to extract an initial rate. However here, the results presented in figure 9 are quite strange. The initial slopes of ln (A/A0) vs time seem similar for 2.5 and 10 µg/mL Ag-pABA-TiO2 catalysts (maybe even slightly smaller in absolute value for 10 µg/mL than 2.5 µg/mL) while for 5 µg/mL Ag-pABA-TiO2 the signal decreases much faster with roughly a factor of 10. A latency is observed in the case of 10 µg/mL, and not for the other amounts. Is that result reproducible? If yes, how explain it.
Author Response
Please find the answers in the attachment
